# Genome-Wide Analysis of Nubian Ibex Reveals Candidate Positively Selected Genes That Contribute to Its Adaptation to the Desert Environment

**DOI:** 10.3390/ani10112181

**Published:** 2020-11-22

**Authors:** Vivien J. Chebii, Samuel O. Oyola, Antoinette Kotze, Jean-Baka Domelevo Entfellner, J. Musembi Mutuku, Morris Agaba

**Affiliations:** 1School of Life Science and Bioengineering, Nelson Mandela Africa Institution of Science and Technology, P.O. Box 447, Arusha 23306, Tanzania; morris.agaba@nm-aist.ac.tz; 2Biosciences Eastern and Central Africa—International Livestock Research Institute (BecA-ILRI) Hub, P.O. Box 30709, Nairobi 00100, Kenya; j.domelevoentfellner@cgiar.org (J.-B.D.E.); josiah.mutuku@wave-center.org (J.M.M.); 3International Livestock Research Institute (ILRI), P.O. Box 30709, Nairobi 00100, Kenya; s.oyola@cgiar.org; 4South African National Biodiversity Institute, Pretoria, P.O. Box 754, Pretoria 0001, South Africa; A.Kotze@sanbi.org.za; 5Department of Genetics, University of the Free State, P.O. Box 339, Bloemfontein 9300, South Africa; 6Current Address: Central and West African Virus Epidemiology (WAVE), Pôle Scientifique et d’Innovation de Bingerville, Université Félix Houphouët-Boigny, Abidjan 01 BP V34, Cote d’Ivoire

**Keywords:** *Capra nubiana* genome, positive selection, desert adaptation, dN/dS analysis, solar radiation

## Abstract

**Simple Summary:**

The Nubian ibex is a wild relative of the domestic goat found in hot deserts of Northern Africa and Arabia. The domestic goat is an important livestock species that is mainly found in arid and semi-arid regions of Africa and Asia. The Nubian ibex is well adapted to challenging environments in hot deserts characterized by high diurnal temperatures, intense solar radiation, and scarce water resources. It is therefore important to understand the genetic basis of its adaptation for scientific and economic importance. To identify genes with adaptive traits, the Nubian ibex genome was sequenced and compared with that of related mammals. We identified twenty-five genes under selection in the Nubian ibex that play diverse biological roles such as immune response, visual development, signal transduction, and reproduction. Three other genes under adaptive evolution involved in protective functions of the skin against damaging solar radiation in the desert were identified in Nubian ibex genome. Our finding provides valuable genomic insights into the adaptation of Nubian ibex to desert environments. The genomic information generated in this study can be used in developing appropriate breeding programs aimed at enhancing adaptation of local goats to less favorable habitats in response to changing climates.

**Abstract:**

The domestic goat (*Capra hircus*) is an important livestock species with a geographic range spanning all continents, including arid and semi-arid regions of Africa and Asia. The Nubian ibex (*Capra nubiana*), a wild relative of the domestic goat inhabiting the hot deserts of Northern Africa and the Arabian Peninsula, is well-adapted to challenging environments in hot deserts characterized by intense solar radiation, thermal extremes, and scarce water resources. The economic importance of *C*. *hircus* breeds, as well as the current trends of global warming, highlights the need to understand the genetic basis of adaptation of *C*. *nubiana* to the desert environments. In this study, the genome of a *C*. *nubiana* individual was sequenced at an average of 37x coverage. Positively selected genes were identified by comparing protein-coding DNA sequences of *C*. *nubiana* and related species using dN/dS statistics. A total of twenty-two positively selected genes involved in diverse biological functions such as immune response, protein ubiquitination, olfactory transduction, and visual development were identified. In total, three of the twenty-two positively selected genes are involved in skin barrier development and function (ATP binding cassette subfamily A member 12, Achaete-scute family bHLH transcription factor 4, and UV stimulated scaffold protein A), suggesting that *C*. *nubiana* has evolved skin protection strategies against the damaging solar radiations that prevail in deserts. The positive selection signatures identified here provide new insights into the potential adaptive mechanisms to hot deserts in *C*. *nubiana*.

## 1. Introduction

The Nubian ibex (*Capra nubiana*), is one of the nine species of the *Capra* genus, which also includes the domesticated goat (*Capra hircus*) [1]. *Capra* species inhabit diverse environments ranging from extreme cold deserts of Siberia (*Capra sibirica*), through the rugged high altitude ranges in Ethiopia (*Capra walie*), and moderate environments of the Zagros mountains (*Capra aegagrus*) to extremely hot deserts of northern Africa and Arabia (*Capra nubiana*) [2]. *C*. *nubiana* population is estimated to be approximately 2500 and is categorized as Vulnerable on the International Union for Conservation of Nature Red List [3,4]. *C*. *nubiana* thrives well in challenging environmental conditions characterized by high diurnal temperatures, intense solar radiation, and limited water supply. In contrast, the domestic goat is a more versatile species that is found in all major agro-ecological zones in Africa and Asia. While *C*. *nubiana* is adapted to harsh desert environments, a large proportion of the domestic goats occur in semi-arid zones where the climate is expected to become hotter and drier as predicted by climate change models [5]. *C*. *nubiana* has adaptive phenotypes such as a shiny waterproof coat that reflects harsh sunlight and minimizes water losses through the skin [2]. The genetic basis behind *C*. *nubiana’s* adaptation to its environment, however, need to be identified, and could translate into useful information for breeding programs in livestock species in the context of the global effects of climate change.

The genetic basis of adaptation is detectable using genomic data through the comparison of sequence data from the target species with that of a suitable reference. A genome-wide comparison of the relative rate of nonsynonymous (dN) versus synonymous substitutions (dS) in protein-coding genes, with the ratio denoted as ω = dN/dS, is an established approach for detecting adaptive evolution processes of a given species among its “peer” [6,7,8]. A ratio ω < 1 indicates negative (purifying) selection, ω = 1 indicates neutral evolution, while ω > 1 indicates positive selection (adaptive evolution) driving the fixation of amino acid changes [9,10]. The dN/dS statistic was initially developed to detect positive selection in individual genes; however, it has been scaled up to detect selection in protein-coding genes at a whole-genome level [10,11]. The dN/dS statistic has been used with success to detect positively selected genes in several species such as viruses [12,13], bacteria [14,15], plants [16,17,18], and higher vertebrates [19,20,21,22,23,24].

The dN/dS statistics is a robust computational tool for detecting protein evolution in genomes with a good correlation with experimental methods such as gene expression studies [25,26]. Protein evolution studies in highland fish, *Gymnocypris przewalskii*, showed that a set of adaptive immune system genes were under positive selection, with high expression after parasitic infection [25]. Similarly, positively selected genes involved in feeding habits in tsetse flies were shown to be highly expressed in organs associated with feeding success such as salivary glands and midgut [26].

The maximum likelihood method based on comparative genomics implemented in the PAML package is widely used to estimate the dN/dS ratio as a measure of protein evolution [27]. Protein evolution analyses using dN/dS statistics have provided clues into the diverse adaptations seen in mammalian species [19,28,29]. For instance, a genome-wide comparison between giraffe and okapi showed that bone development genes (Fibroblast Growth Factor Receptor-like 1 and Notch Receptor 4) were under positive selection in the giraffe, and are likely associated with the elongated body structure [28]. The giraffes’ distinct stature and body morphology are thought to be a feeding adaptation, which enables it to feed on tall acacia trees in savanna landscapes [28]. Genome sequence comparison of dromedary and bactrian camels with alpaca showed that camels have evolved adaptive mechanisms to cope with environmental stresses in deserts as evidenced by positive selection of oxidative stress response genes (Endoplasmic Reticulum Protein 44 and Microsomal glutathione S-transferase 2) [19]. Similarly, genome-wide comparisons of fifty-four ruminants showed that bovids which inhabits grasslands displayed signals of positive selection in cursorial locomotion genes (angiotensin I converting enzyme and erythropoietin) which are important for endurance [21]. Comparative genomic sequence analysis of reindeer with nine other mammalian species showed that vitamin D metabolism genes (Cytochrome P450 Family 27 Subfamily B Member 1 and Cytochrome P450 oxidoreductase) were under selection [30]. Selection of vitamin D metabolism genes is an adaptive mechanism that enables reindeer to produce high levels of vitamin D needed for calcium absorption and body fat oxidation, which are required to survive in Arctic environments [30].

The objective of this study was to identify candidate protein-coding genes that can underlie the adaptation of *C*. *nubiana* to its desert environment.

## 2. Materials and Methods

### 2.1. Samples

The animal sample used in this study was obtained from the National Zoological Garden, Pretoria, South Africa. A liver tissue sample collected postmortem from a seven month old female *C*. *nubiana* that died of natural causes was requested from the National Zoological Garden biobank. Additional information of the *C*. *nubiana* used in this study is provided in Appendix A. Genomic DNA was isolated using the phenol-chloroform extraction method. The extracted DNA was quantified using the Nanodrop 2000C spectrophotometer (Thermo Fisher Scientific Inc. Woltham, Ma, USA), and the quality was assessed using electrophoresis in 1.5% agarose gel. The National Zoological Gardens approved all animal procedures for tissue sampling (Ethical clearance number: NZG/P14/13).

### 2.2. Sequence Data Generation

Approximately 200 ng of the purified genomic DNA was used to construct a library of insert size of 450 bp using TruSeq Nano library prep kit following the manufacturers’ protocol (Illumina, San Diego, CA, USA). The library was sequenced on an Illumina HiSeq 2500 platform in High Output mode using a Hiseq SBS kit V4. Library to produce paired-end sequence with a 125 bp read length. Library preparation and sequencing were done at the Agricultural Research Council’s Biotechnology Platform (ARC-BTP) based at the Onderstepoort Veterinary Institute campus, Pretoria, South Africa. The quality of the raw sequence reads were assessed using FastQC version 0.10.065 [31]. Adapters, PCR duplicates, and overrepresented sequences were trimmed off using Trimmomatic version 0.32 [32]. The genome size was predicted from the trimmed paired-end sequence reads using Kmergenie version 1.7016 [33].

### 2.3. Identification of Single Nucleotide Variants (SNVs) between C. Nubiana and the Domestic Goat

Paired-end sequence reads that passed the quality control assessment were aligned to the domestic goat reference genome (ARS1 assembly: GCA_001704415.1) [34] using Burrows–Wheeler Alignment Maximal Exact Match algorithm (BWA-MEM) version 0.7.15 [35] using default. SAMtools was used to convert the Sequence Alignment Map (SAM) file into indexed and sorted Binary Alignment/Map (BAM) format [36]. Single nucleotide variant calls against the domestic goat reference genome were generated using SAMtools mpileup [36] with parameters set to *-q 30 -Q 30*, where *-q 30* sets the minimum mapping quality and *Q 30* sets the minimum base quality. The *mpileup* output file (Binary Call Format (BCF)) file, was redirected to the BCFtools *view* program to convert it to Variant Call Format (VCF) format [36]. The variant calls were then filtered using vcfutils.pl varFilter with the minimum and maximum read depths set to 6 and 100 reads, respectively [36]. The transition-to-transversion (Ti/Tv) ratio, a parameter used to assess the specificity of new SNP calls [37], was estimated using vcftools version 0.1.15 [38]. Functional annotations of the SNVs were performed using Variant Effect Predictor (VEP) tool version 96 [39] with *Capra hircus* genome (ARS1 assembly: GCA_001704415.1) as the reference [34].

### 2.4. Capra Nubiana and Capra Hircus Protein-Coding DNA Sequences (CDS)

The *C. nubiana* genome assembled from the Illumina short reads in this study was highly fragmented; hence the protein-coding DNA sequences were generated from the domestic goat coding DNA sequences. All CDS for Capra hircus genome assembly GCA_001704415.1 downloaded from Ensembl BioMart [40] were used as template to generate corresponding *C. nubiana* gene models. Briefly, a custom-made bash script was used to replace nucleotides in the coding DNA sequences of the domestic goat assembly with the corresponding *C. nubiana* alleles based on the homozygous coding SNVs positions identified in the variant calling pipeline above (Section 2.3). Visual inspections of randomly selected *C. nubiana* CDS were carried out by comparing it side by side with the corresponding domestic goat CDS. The visual inspection confirmed that the domestic goat allele at the SNVs positions were successfully replaced with *C. nubiana* alleles. The bash script, coding DNA sequences of the domestic goat, *C. nubiana* and, SNVs annotation file used are provided at Figshare (https://figshare.com/s/36c4effaa8d50c08f0f7).

### 2.5. Protein-Coding DNA Sequences (CDS) for Positive Selection Analysis

Detection of positive selection signatures using branch-site model requires that the branches in a phylogeny tree is partitioned into foreground and background branches. It is expected that the foreground branches have sites evolving under positive selection, while background branches will have sites evolving under negative, purifying, or natural selection. The protein-coding DNA sequences for *C. nubiana* were generated from the domestic goat CDS as described in Section 2.4. All CDS for each of the background species (domestic goat, cattle, sheep, wild yak, American bison, horse, donkey, tiger, cat, dog, pig, and panda) were downloaded from Ensembl v.97 [41]. The Tibetan antelope and water buffalo CDS were obtained from the genome data, downloaded from Genbank [42]. The data source for each of the taxa is provided in Appendix A.

### 2.6. Single Gene Ortholog Identification

The assembled CDS of *C*. *nubiana*, *C*. *hircus*, and the background species were used to identify single-copy gene orthologs. The single-copy gene orthologs shared among the 15 species were identified using reciprocal best hit (RBH) approach implemented using blastn [43,44] with parameters set to: e-value of 1 × 10^−10^, coverage > 70%, and percentage identity > 50%. Pairwise orthologs were derived between *C*. *nubiana* CDS and each of the 14 species and the intersection across all the pairs were used to construct a combined single-copy gene set. A gene pair was considered an ortholog if they appeared as the best hits of each other in the pairwise homology search. Single-copy gene set present in at least seven species including the core species (*C*. *hircus* and *C*. *nubiana*) was retained for subsequent analysis.

### 2.7. The dN/dS Analysis

The CDS of the single-copy gene orthologs were translated to the corresponding polypeptides using the mod_translate program [45], and any sequence with internal stop codons was discarded. The polypeptides sequences were aligned using the MUSCLE program version 3.8.1551 [46], and the resulting alignments were used to guide coding sequence alignments using the RevTrans program version 1.4 [45]. The CDS alignments were used to construct phylogeny trees using the PhyML package, version 3.0 [47]. *C*. *nubiana* leaf in each of the phylogeny tree was labeled as the foreground branch, while the other species were set as background using the ETE toolkit, version 3.1.2 [48].

Based on the sequence alignment of each gene set of the single-copy gene orthologs and the corresponding phylogenetic tree, dN/dS analysis was carried out using revised branch-site model A [49] implemented in CodeML program of the PAML package, version 4.7a [50]. The CodeML parameters used are provided in Appendix A. The genes with significant *p*-values (<0.05) based on Likelihood Ratio Test (LRT) *X*^2^-analysis were considered to be under adaptive evolution and were selected as the initial positively selected genes (PSGs) list. Since the branch-site model is sensitive to the taxa sample size [51], the initial candidate PSGs were re-analyzed after adding more data to the core set such that each gene set had a minimum of ten and a maximum of nineteen sequences. The additional CDS corresponding to each PSG used for re-analysis were obtained from even-toe ungulates from which sequences were available in public databases. Furthermore, amino acid sites of the final candidate genes under positive selection were identified using the Bayes Empirical Bayes (BEB) algorithm [52]. A site was considered to be positively selected when the posterior probability was greater than 80% [52]. Additional paired-end sequence reads for two *C*. *nubiana* individuals were downloaded from the National Center for Biotechnology Information database (https://www.ncbi.nlm.nih.gov/) under Sequence Read Archive (SRA) accession number SRR8437789 and SRR8437792 and analyzed following a similar approach used in Section 2.3. The animal samples for the additional two *C*. *nubiana* individuals were obtained from Egypt and Saudi Arabia [53]. SNV sites across the three *C*. *nubiana* individuals were extracted using a bash script.

### 2.8. Functional Annotation of the Positively Selected Genes (PSG) and Sites

The gene ontology (GO) terms assignments for the PSGs were found by searching the genes in Ensembl Goat Genes v.97 using Biomart [40], while additional gene functions were sourced from the literature. Gene enrichment analysis was carried out using The Database for Annotation, Visualization and Integrated Discovery (DAVID) version 6.8 [54]. Furthermore, functional impact analysis of the amino acid substitutions in positively selected sites of the candidate genes was carried out using Polyphen-2 (Polymorphism Phenotyping-2) [55]. Polyphen-2 was run with default cutoff values. Amino acid substitutions with score < 0.2 were considered to be benign, scores between 0.2–0.85 were considered as a possibly damaging variant, while scores between 0.85–1 were considered as probably damaging.

## 3. Results

### 3.1. Genome Sequence and SNVs Calling

The *C*. *nubiana* genome sequence was determined by constructing paired-end libraries followed by sequencing using Illumina Hiseq 2500 yielding approximately 900 million raw reads. A total of 781 million paired-end sequence reads were retained after quality control analysis, representing ~37x sequence coverage of the estimated 2.63 Gbp *C*. *nubiana* genome. The clean paired-end sequence reads mapped to approximately 98% of the domestic goat reference genome (ARS1 genome version). Genome sequence comparison of *C*. *nubiana* and the domestic goat yielded a total of 19,468,467 SNVs sites; 16,443,766 of them were homozygous SNVs. Most of the SNVs were located in the non-coding regions of the genome (intergenic: 69.2%, intronic: 29.6%), and the remaining small percentage (0.7%) were located in the exonic regions. The alignment file and SNVs data are provided in Figshare https://figshare.com/s/3041e34bc83934ba5797.

### 3.2. Positively Selected Genes in Capra Nubiana

Homozygous *C*. *nubiana* SNVs alleles were projected to *C*. *hircus* CDS to generate a total of 19,418 *C*. *nubiana* CDS. Subsequent orthologs identification yielded a total of 15,527 single-copy gene orthologs shared by *C*. *nubiana*, *C*. *hircus*, and at least seven of the fifteen selected background taxa. The initial dN/dS analysis using a minimum of seven and a maximum of fifteen species as the background data showed that 34 genes were under positive selection in *C*. *nubiana*. Using additional background taxa data (minimum of ten and maximum of nineteen species), we confirmed 28 out of the initial 34 candidate genes to be under positive selection in *C*. *nubiana*. Approximately 98% of the SNV sites shown to be under positive selection in 22 genes were consistent across three *C*. *nubiana* individuals. The BEB analysis showed that 42 amino acid sites in the 22 candidate genes were under selection.

Functional impact analysis of amino acid substitutions conducted using Polyphen-2 at the sites identified as positively selected by BEB algorithm in the candidate genes showed that 17 amino acid changes were classified as “possibly damaging” or “probably damaging,” while 13 were classified as “benign.” The possibly damaging or probably damaging amino acid substitutions are likely to alter the protein structure and function. Positively selected gene list, sites, and the functional impact of the corresponding amino acid substitutions are provided in Table 1 and Appendix A.

The gene ontology (GO) assignments showed that the positively selected genes are involved in diverse molecular functions such as protein binding, ATP binding, olfactory receptor activity, serine-type endopeptidase activity, metal ion binding, and G protein-coupled receptor activity. Additionally, other positively selected genes are involved in biological processes that include: camera-type eye development, prostaglandin metabolic processes, signal transduction, G protein-coupled receptor signaling pathway, transmembrane transport, protein ubiquitination, DNA replication, positive regulation of Notch signaling pathway, negative regulation of systemic arterial blood pressure, spermatogenesis and oocyte development were identified. The gene ontology terms are provided in Appendix A. Additionally, we found positively selected genes such as ATP binding cassette subfamily A member 12 (*ABCA12*), Achaete-scute family bHLH transcription factor 4 (*ASCL4*), and UV stimulated scaffold protein A (*UVSSA*) that are involved in 10 GO biological processes such as keratinization, ceramide transport, the establishment of the skin barrier, lipid transport, skin development, transcription-coupled nucleotide-excision repair, and response to ultra-violet (UV) radiation, which may play different roles in desert environment adaptations.

### 3.3. Skin Development and Barrier Function Genes under Positive Selection in C. Nubiana

In total, three of the twenty-two positively selected genes identified in this study are involved in skin barrier development and functions (*ABCA12*, *ASCL4,* and *UVSSA*). The *ABCA12* gene, a member of ATP-binding cassette (ABC) transporters is found in chromosome 2 of the domestic goat (ARS1 assembly). The *ABCA12* gene had one amino acid substitution (M570T) with a BEB posterior probability of 94% classified as ‘possibly damaging’ (Polyphen-2 score of 0.74). The positively selected site in *ABCA12* is outside the known functional domains for this gene. Gene ontology (GO) terms associated with *ABCA12* include lipid transport activity, keratinocyte differentiation, ceramides transport, surfactant homeostasis, and establishment of skin barrier. An illustration of the gene tree and multiple sequence alignment data used for positive selection analysis of *ABCA12* is provided in Figure 1.

Achaete-scute family (basic helix-loop-helix) bHLH transcription factor 4 (*ASCL4*) is a transcriptional regulatory protein found in chromosome 5 of the domestic goat (ARS1 assembly). The *ASCL4* gene had one amino acid substitution (S30L) with a BEB posterior probability of 99.9%, which is not within the known gene functional domain. Functional impact analysis of the amino acid substitution (S30L) predicted it to be ‘probably damaging’ (Polyphen-2 score of 0.99). The *ASCL4* gene is one of the five homologs of drosophila Achaete-Scute basic helix-loop-helix (bHLH) transcription factors (*ASCL1*, *ASCL2*, *ASCL3*, and *ASCL5*) [54]. The *ASCL1* and *ASCL2* genes are involved in the development and differentiation of neural crest cells in the sympathetic system, *ASCL3* is involved in the development of the duct cells in salivary glands and *ASCL5* is expressed in the brain, though its function is yet to be determined [56,57]. GO terms associated with *ASCL4* include; transcription, regulation of transcription from RNA polymerase II promoter, and skin development. An illustration of the gene tree and multiple sequence alignment data used for positive selection analysis of *ASCL4* is provided in Appendix A.

UV-stimulated scaffold protein A (*UVSSA*), a DNA repair gene found in chromosome 6 of the domestic goat (ARS1) assembly, had two amino acid substitutions (D361G and A517T). The amino acid change at position 361 had a BEB posterior probability of 91.7% classified as ‘probably damaging’ (Polyphen-2 score of 0.992). While the mutation at position 517 had a BEB posterior probability of 89.7 classified as benign. The amino acid replacement at position 361 of *UVSSA* gene is located in the DUF2043 domain. Gene ontology terms associated with the *UVSSA* gene include transcription-coupled nucleotide-excision repair, response to UV, and protein ubiquitination. An illustration of the gene tree and multiple sequence alignment data used for positive selection analysis of *UVSSA* is provided in Appendix A.

## 4. Discussion

### 4.1. Whole-Genome Mapping, Single Nucleotide Variant Calling, and Annotation

The *C*. *nubiana* genome was sequenced to a depth of 37x, a sufficient coverage recommended for SNVs detections [58]. Mapping of *C*. *nubiana* sequence reads to the domestic goat reference genome showed that 98% of the reads mapped to unique sites; an indication of high-quality sequence data for detections of genetic variants [58].Approximately, 19 million SNVs identified in this study are comparable to the number identified in other interspecies studies; for instance, a total of 18.2 million SNVs were detected by comparing donkey with horse genome [59]. The SNVs transition to transversion (Ts/Tv) ratio was 2.39 which is close to the empirical human Ts/Tv ratio (>2.1), this indicates a relatively low potential random sequencing errors [37,60].

The analysis was based on one individual *C*. *nubiana* and the domestic goat (*C*. *hircus*), which acted as a proxy for the respective species. A key presumption made in this study was that the differences between the species reflect the 2.85 million years of divergence, and they are more likely to be fixed in the respective species [61]. Inevitably, some of the SNVs discovered here might be linked to that individual *C*. *nubiana* and the domestic goat from whom the ARS1 assembly was developed; however, we are confident that the majority of the SNVs identified reflect species–specific fixed variations. Validation of SNVs showed that 98% of the SNV sites shown to be under positive selection in 22 genes were consistent across the three *C*. *nubiana* individuals, thus confirming our assertation that the majority of the selection signals detected in this study reflect species-specific fixed variations.

### 4.2. Positively Selected Genes in Capra Nubiana

A total of twenty-two genes involved in diverse biological functions were shown to be under positive selection in *C*. *nubiana*. A total of nineteen of the positively selected genes are involved in visual development (Serine protease 56 and ATP binding cassette subfamily B member 5), blood pressure regulation (Prostaglandin I2 synthase and Rho GTPase activating protein 42), reproduction (Meiosis Specific With Coiled-Coil Domain, Storkhead box 2 and Eukaryotic translation initiation factor 2 subunit beta), and ion transport (Atpase H+ transporting V1 subunit E2 and Matrix AAA peptidase interacting protein 1). In addition, genes involved in signal transduction (Olfactory receptor 2G2-like, Olfactory receptor 1P1 and Putative olfactory receptor 52P1), protein ubiquitination (F-box protein 21), regulation of Notch signaling pathway (Nucleolus and neural progenitor protein), angiogenesis (Multimerin 2), and phagocytosis (Toll-like receptor adaptor molecule 2) were shown to be under selection in *C*. *nubiana*. The functional roles for most of the positively selected genes identified in *C*. *nubiana* are less clear. Further studies need to be carried out to delineate their possible adaptive roles. However, three genes *ABCA12*, *ASCL4,* and *UVSSA* involved in skin barrier development and function, which may have a role in adaptations to desert environments, were shown to be under positive selection in *C*. *nubiana*.

### 4.3. Skin Development and Barrier Function Genes under Positive Selection in C. Nubiana

*C*. *nubiana* is exposed to high diurnal temperatures and intense solar radiation that are likely to increase the rate of water loss through the skin or induce skin damages. This implies that excellent epidermal barrier system is needed to minimize water losses through the skin as reflected by *C*. *nubianas*’ shiny waterproof coat [2]. Genes involved in skin barrier development such as *ABCA12* and *ASCL4* displayed strong selection signals in *C*. *nubiana*. *ABCA12* is a keratinocyte transmembrane protein that transports lipids and ceramides via lamellar granules, which form the skin–lipid barrier in the stratum corneum [62]. The skin barrier provides protection against solar radiation, water loss, and pathogens [63]. Mutations within *ABCA12* conserved domains are linked to skin disorders known as ichthyosis, a condition whereby patients are unable to accumulate lipids in the stratum corneum, hence are exposed to life-threatening water loss through the skin [64,65] Functional impact analysis of the amino acid substitution in *ABCA12* showed that the change is likely to alter the protein structure and functions. We hypothesize that these functional changes maybe the genetic basis behind *C*. *nubiana* adaptation to its desert environment.

Similarly, *ASCL4* a basic helix-loop-helix (bHLH) protein was under selection in *C*. *nubiana*. Basic helix-loop-helix (bHLH) proteins including the *ASCL4* gene play key roles in nervous system development; however, recent evidence has shown that they are also important in epidermal development [66]. The function of *ASCL4* is not known, but its expression is restricted to the skin and especially the fetal skin [67] where it may, among other roles, be involved in the development and growth of hair follicles [68]. Functional impact analysis of the amino acid substitution in the *ASCL4* gene showed that the change is likely to alter the protein structure and functions. The identification of *ABCA12* and *ASCL4* as positively selected in *C*. *nubiana* provides evidence of their possible roles in the skin barrier development, where they may be of significance in adaptation to hot desert conditions.

In addition to an elaborate skin barrier, *C*. *nubiana* has evolved genetic mechanisms in response to the damaging ultraviolet radiations (UV) in the desert. In this study, we identified a DNA repair gene (*UVSSA*) putatively involved in protecting *C*. *nubiana* from the damaging desert solar radiation to be under positive selection. The *UVSSA* gene removes impaired DNA located in actively transcribed genes in response to UV damage [69]. Mutations in the *UVSAA* gene are linked to UV-sensitive syndrome in humans, and impaired transcription-coupled nucleotide-excision repair system [70]. We suggest that *UVSSA* in *C*. *nubiana* may have a role in repairing DNA damages induced by intense solar radiation in the hot deserts.

## 5. Conclusions

This study showed that comparative analysis of protein-coding genes is a robust method for detecting signals of selection in genomes. A total of twenty-two genes that play diverse biological roles in *C*. *nubiana* were identified to be under positive selection. In total, three out of the twenty-two genes (*ABCA12*, *ASCL4*, and *UVSSA*) are involved in skin barrier development and function. Therefore, we conclude that *C*. *nubiana* has evolved skin protection strategies to minimize water losses and the damaging effects of solar radiation in the hot desert habitats where it thrives. The study further demonstrated that a comparison of wild relatives of the domestic goat is useful for identifying candidate genes that can be used in breeding programs aimed at improving the domestic goat to adapt to challenging environments. The results of this study are limited the by use of few individuals, hence the candidate genes identified in this study need further confirmation through empirical studies to delineate their possible roles in adaptations. The identification of key genes involved in the adaptation to the desert environment in *C*. *nubiana* may have applications in breeding programs, and form a valuable genomic resource for further adaptive evolution studies in *Capra* species.

## Figures and Tables

**Figure 1 animals-10-02181-f001:**
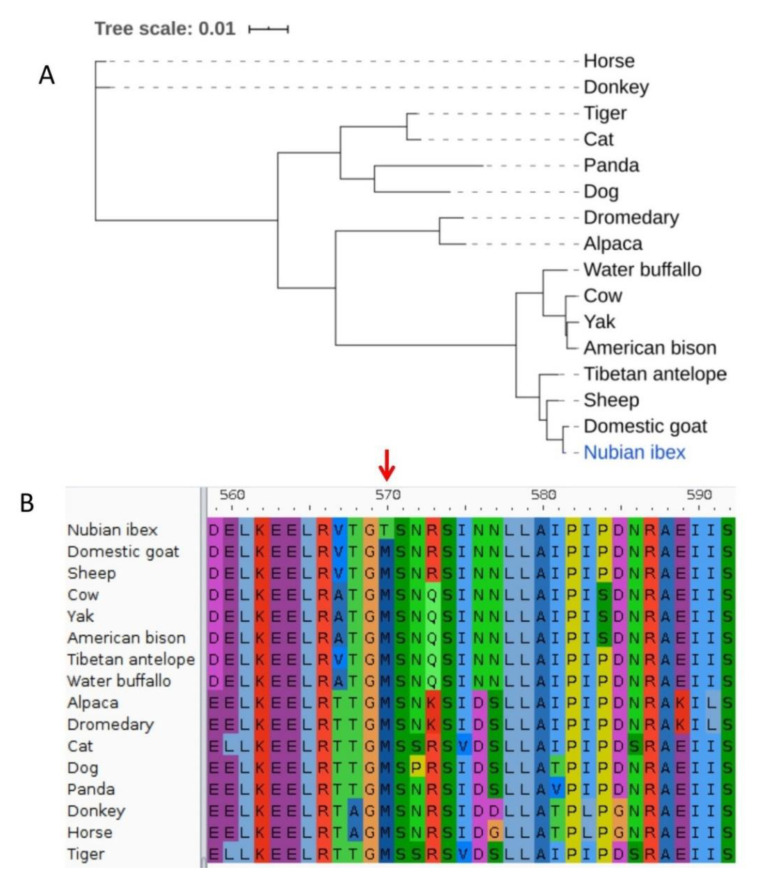
An illustration of the ATP binding cassette subfamily A member 12 (*ABCA12*) phylogeny tree and alignment data used for dN/dS analysis. (**A**) The maximum likelihood phylogenetic tree from *ABCA12* amino acid sequences of 16 species used for dN/dS analysis. (**B**) The multiple sequence alignment of the 16 species used of dN/dS analysis showing amino acid site (M570T) under positive selection in *ABCA12* gene.

**Table 1 animals-10-02181-t001:** Positively selected genes in *C*. *nubiana*.

Ensembl Gene Id	Gene Name	Positively Selected Sites with BEB Posterior Probability > 0.8	Polyphen-2 Functional Impact Analysis
ENSCHIT00000003090	Storkhead box 2	T734V, N835T	Benign
ENSCHIT00000004084	Atpase H+ transporting V1 subunit E2	M72N	Possibly damaging
ENSCHIT00000004434	Olfactory receptor 2G2-like	F73T	Probably damaging
ENSCHIT00000008957	Serine protease 56	Q424L, R425G, R436W	Benign, probably damaging, benign
ENSCHIT00000010253	Matrix AAA peptidase interacting protein 1	T76A, Q93P	Benign
ENSCHIT00000012782	Putative olfactory receptor 52P1	M67L	Possibly damaging
ENSCHIT00000015750	Prostaglandin I2 synthase	R320H, D411E	Possibly damaging, benign
ENSCHIT00000018881	F-box protein 21	S603A, E606G, K615E, E620G	Benign
		K616R	Possibly damaging
ENSCHIT00000026283	Zinc finger and SCAN domain containing 23	P213N	Probably damaging
ENSCHIT00000028977	UV stimulated scaffold protein A	D361G, A517T	Probably damaging benign
		E99Q	Probably damaging
ENSCHIT00000030384	F-box and WD repeat domain containing 2	L82C	Probably damaging
ENSCHIT00000000612	Multimerin 2	S214H	Probably damaging
ENSCHIT00000015914	Toll like receptor adaptor molecule 2	I213N	Benign
ENSCHIT00000016318	Eukaryotic translation initiation factor 2 subunit beta	K83I, K205E	Possibly damaging
ENSCHIT00000020934	LY6/PLAUR domain containing 6B	A7T, F16L	Benign
ENSCHIT00000028741	ATP binding cassette subfamily A member 12	M570T	Possibly damaging
ENSCHIT00000035903	PATJ crumbs cell polarity complex component	V249I, I1739V	Benign
		I1738F	Probably damaging
ENSCHIT00000036547	Rho gtpase activating protein 42	I502L, M770T	Benign
		W773R	Probably damaging
ENSCHIT00000040177	Achaete-scute family bhlh transcription factor 4	L30S	Probably damaging
ENSCHIT00000040379	Olfactory receptor 1P1	A133T	Benign
		V135D, H159C	Possibly damaging
ENSCHIT00000034768	Tripartite motif containing 16	D159L, S515L	Benign
ENSCHIT00000041152	Centrosomal protein 112	K338G	Unknown

Posterior probabilities were obtained from Bayes Empirical Bayes (BEB) analysis. The positively selected sites column shows the position of the amino acid substitutions in respective genes; where T for example in (T734V) represent the ancestral amino acid, 734 indicates the position, while V is the *C*. *nubiana* amino acid.

## Data Availability

The sequence data (FASTQ) files used in this study have been deposited in to the National Centre of Biotechnology Information (NCBI) under Bioproject accession number PRJNA674751.

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
