# Peer review of "Genome-Wide Analysis of Nubian Ibex Reveals Candidate Positively Selected Genes That Contribute to Its Adaptation to the Desert Environment"

_animals, 2020, doi:10.3390/ani10112181_

Round 1

Reviewer 1 Report

Dear authors,

the manuscript entitled “” deals with the comparison between Nubian Ibex and other livestock species genomes.

Even if the aim of the manuscript could be of interest, the manuscript needs some improvements before to be considered for publication.

My main concerns are:

1) The “materials and methods” section is quite hard to follow, because it contains lots of information written in a quite confusing way (also because of English problems, please see the comment below). Moreover, some important details (e.g., number of samples obtain from National Zoological Garden, number of sequences downloaded…) are missing in this section. If you really used just one animal (line 116 and 306), can we be sure about the results and their repeatability? I don’t strongly support the idea presented at lines 309-312 that your results “reflect species-specific fixed variations”. Can you please add something to support this conclusion? And please strengthen this concept also in conclusion (add something at lines 377-378).

2) English language should be revised by a native person throughout the whole text. For example, please see line 61: “… it is estimated that here are 2500 mature individuals in the wild” is not quite English. The sentence at lines 73-75 starting with “A genome-wide….” makes no sense, you are missing a verb or something else.

3) Please rewrite all references as reported in the authors guidelines:

Author 1, A.B.; Author 2, C.D. Title of the article. Abbreviated Journal Name Year, Volume, page range.

Line-by-line points

Lines 18-19: please replace “…mainly found in…” with “…mainly raised in…”

Line 38: please change “to its…” with “to the…”

Lines 42-51: this paragraph has a bigger font size

Line 52: please remove green color

Line 85: please add a comma after the italics name and remove “for instance” (wrong place)

Lines 110-112: you cannot say “our results are of importance….” in the aim of the manuscript. You did not say anything about results at this point.

Line 115: How many samples did you obtain from National Zoological Garden? Did you ask for only one “individual female C. nubiana? Why is important to know that the animal “died of natural causes”? At line 306 you said that the study was “based on one individual C. nubiana”. If you used just one individual of C. nubiana, why did you write samples at line 115 and 389?

Lines 142-144: please add a comma after “(TI/TV)” and after “[37]”.

Line 148: How many “sequences” did you download?

Lines 161-164 and 166-168: why did you download all these species to analyze a goat relative? I mean, I can understand the use of other livestock species (such as cattle, sheep, yak, bison, horse, donkey and buffalo), but which is the reason of using tiger, cat, dog and panda?

Line 170: please remove the semicolon or replace with “:”

Lines 222-223: you already reported this (“15,27 single-copy gene orthologs”) in Materials and Method.

Lines 233-234: please be careful in the use of semicolons “…that include;” “processes;”

 Best wishes

Reviewer 2 Report

The manuscript might report scientifically important findings. However, English and the style must be improved. Therefore, I recommend the resubmission after careful English proofreading. For example...

There is no information about Affiliation No. 2.

The name of only the forth author was in italics.

The style of abstract was strange.

The style of references were not suitable for this journal at this time.

And so on...

Reviewer 3 Report

The manuscript entitled “Genome-Wide Analysis of Nubian Ibex Reveals Candidate Positively Selected Genes that Contribute to its Adaptations to the Desert Environment”. This study used dN/dS statistic to detect the positively between C. nubiana and related species and identified many genes are under positive selection. Several genes are involved in skin barrier development and function, suggesting the C. nubiana has evolved skin protection strategies against the damaging solar radiation in deserts. The study is well designed, and I only have a few minor suggestions and comments.

  1. Abstract: “The Nubian ibex (Capra nubiana)………”in line 42-51, and the abstract should be rephrased.
  2. Line 50, “generated in this study can be used to improve the domestic goats to adapt to less productive”, this sentence is difficult to understand.
  3. The result of multiple sequence alignment may due to the contingency. It’s better for you to do a analysis if hot deserts animals were conservative in this loci compare with the different environment.
  4. Please use the scientific notation according the requirement of Journal. For instance, Line 170 e-value 1e-10.
  5. Line 146, it’s better to offer the website of annotation information about Capra hircus.
  6. Table 1, “Ensemble” > “Ensembl”. Also, more descriptions can be presented for legend of Table 1.
  7. Line 200, “Ensemble Goat Genes”> “Ensembl Goat Genes”
  8. Line 240 ABCA12, ASCL4, UVSSA, as a gene name, it should be italic.
  9. The figures should be regenerated in high resolutions.
  10. “Supplemental File 4.” and “Supplemental File 3.” should be in right order.

Round 2

Reviewer 1 Report

Dear authors,

your changes improved the quality of the manuscript. I appreciated your explanation about the use of just one sample and the validation you did using the 2 new samples.

I again recommend you to add in the "conclusions" section a sentence that results are "limited" by the use of just one sample. You can add it near this sentence: "However, the candidate genes identified in this study need further confirmation through empirical studies to delineate their possible roles in adaptations."

Best wishes 

Reviewer 2 Report

I this this paper is likely to be suitable for short communication, because the number of sample sequenced was very few and the detailed information of animals sequenced could not found.
